# Intelligent Generation Method and Sustainable Application of Road Systems in Urban Green Spaces: Taking Jiangnan Gardens as an Example

**DOI:** 10.3390/ijerph20043158

**Published:** 2023-02-10

**Authors:** Lina Yan, Yile Chen, Liang Zheng, Yi Zhang, Xiao Liang, Chun Zhu

**Affiliations:** 1Faculty of Humanities and Arts, Macau University of Science and Technology, Taipa, Macau 999078, China; 2Shanghai Total Architectural Design & Urban Planning Co., Ltd., Shanghai 200433, China; 3Shanghai GOODLINKS International Design Group, Shanghai 200051, China; 4Shanghai Pudong Architectural Design & Research Institute Co., Ltd., Shanghai 200120, China

**Keywords:** urban green space, road intelligent generation, design method, parameterization, genetic algorithm

## Abstract

This paper takes the garden road system as the research object and proposes a method of generating paths for classical gardens based on parametric design. Firstly, by studying the distribution characteristics of roads, the data on the curvature, angle, and view area of roads were collected. Secondly, the obtained data were transferred to the parameterized platform, and a method of intelligent generation was used for calculation. Finally, the road system was optimized by the genetic algorithm for better application in modern landscape design. According to the current situation, the road system plan generated by the algorithm inherits the characteristics of classical garden roads. This method can be applied to the courtyard, the community park, the urban park, and other objects. This research not only identifies the characteristics of landscape cultural heritage, but also produces an innovative, intelligent design tool. It provides new methods for the parameterized inheritance and application of traditional landscape heritage.

## 1. Introduction

### 1.1. Research Background

With a global emphasis on ecological and cultural landscapes, urban parks, and national parks, a series of modern gardening movements have followed in areas as large as scenic spots and national parks, and as small as pocket parks by the street and green spaces in the corners of residential areas. China also attaches great importance to the construction of the urban environment and seeks an urban landscape that has its own characteristics and is suitable for contemporary and future urban development. More people are realizing that traditional gardens need to be incorporated into the construction of contemporary urban spaces as a part of the urban area’s characteristics. Garden expositions at certain celebrations, the community green spaces of overseas Chinese, and antique-style modern gardens can be seen everywhere. In terms of city image, there is a lack of understanding of traditional gardens, and the construction of a large number of antique parks neither incorporated nor conveyed traditional characteristics, destroying the public’s impression of traditional spaces. Chinese classical gardens have their own characteristics. In particular, the idea of Jiangnan gardens that create natural forests in the city has been passed on from past generations. Facing the continued development of the urban landscape today and in the future, our concern is how to capture the design essence of traditional gardens in a way that is suitable for contemporary and future needs.

Gardens are not spaces that exist alone, but are an important part of the urban space system. Among them, the garden road system is an important element of classical gardens. It functions to organize traffic, connect garden landscape elements, divide garden tour space, and guide tour routes. It can even be said that the planning and design of their garden roads are the essences of the classical gardens and the focus of contemporary landscape design planning.

Therefore, we want to identify the characteristics of traditional garden roads by studying the garden road system and then find a way to capture the characteristics of these traditional spaces to make the past serve the present, utilize traditional and excellent design ideas in advanced and modern ways, and adapt these characteristics to current spatial development needs and future scenarios.

### 1.2. Literature Review

#### 1.2.1. Analysis of Design Methods of Sustainable Urban Green Space Landscape

At present, urban green spaces mainly include comprehensive parks, community parks, specialized parks, strip parks, and street green spaces, including any water areas within their boundaries. However, the term does not include subsidiary green land, production green land, protected green land, or other green land. In addition, there are small green spaces in buildings for public use that can also be classified as urban public green spaces. In the past, traditional urban green space design was carried out by landscape architects during on-site surveys, and the design was completed in combination with the needs of society or of the property owners. Presently, common urban green space design methods mainly include the following: using virtual reality (VR) technology to obtain user and expert opinions for the design of green business parks (GBPs) [1]; urban green space design that responds to human needs for recreation, amenities, and environmental quality [2,3,4,5]; the optimal planning of an urban greening system for high-density cities in response to the urban microenvironment based on a genetic algorithm [6]; a method for planning urban green spaces that combines natural solutions and GIS tools [7]; medicinal landscape design based on the theory of the five senses [8]; and urban green space design methods based on disease prevention or suppression [9,10,11,12,13,14,15].

#### 1.2.2. Method and Model Analysis of Road Organization in Jiangnan Classical Gardens

However, with the advancement of modern gardening in China in this century, we have begun to explore the methods and models of Jiangnan classical garden space creation and garden road organization. The Craft of Gardens, Jiangnan Garden Chronicles, Suzhou Classical Gardens, and Jiangnan Gardens, the classic works of Chinese gardening, all mention the study of garden paths [16,17].

In modern times, most of the research has been based on text description, and there have been few quantitative research studies. For example, some scholars used “cognitive imagery” to explore the visual perception of the Lingering Garden in space [18]. Another study used the theory of cognitive narratology to explore the influence of paths in private gardens such as the Garden of Cultivation, the Lingering Garden, and the Master of the Nets Garden in Suzhou, which were established during the Ming and Qing Dynasties [19]. The spatial sequence of the entrance of the Lingering Garden was analyzed with the viewshed analysis theory [20]. The path space of private gardens in the Jiangnan region was studied by using the combination of hierarchical classification analysis, modulus theory, “picture-bottom” theory of Gestalt psychology, and topology theory [21]. In addition, some scholars analyzed garden area, path length, path projection length, and morphological change according to Chinese and Western gardens’ winding-path patterns, and obtained the design methods’ rules [22].

In recent years, scholars have applied advanced ideas and techniques to road research. Research addressing quantification, empirical video analysis technology, geographic information data, and other aspects of the field has increased. For example, the spatial layout and path relationship in the Humble Administrator’s Garden, the Lingering Garden, and the Master of the Nets Garden were determined with quantitative indicators including static line of sight, sight distance, visual field, dynamic path spacing, and scene enclosure [23]. The stagnant point research method based on video analysis technology was used to empirically analyze the stagnation point law of the Master of the Nets Garden and the Lingering Garden [24,25]. Another study carried out the simulation and application of the urban road landscape based on geographic information data [26]. Further, research has been conducted on algorithms and the architecture of software systems for automated natural and anthropogenic landscape generation [27].

#### 1.2.3. Research Frontier Analysis of the Intelligent Design Method

Parametric design is a method of intelligent design that uses parameters to describe the variables to realize the repeatability and adjustability of the design [28]. Although parameter manipulation in parametric design cannot be strictly regarded as a design, the design solution is defined as a controlled design application through parameter scripts and algorithms, so that the parametric design has the variability and fixity of general design behavior [29]. In recent years, parametric design has been widely used. In the field of construction decoration, a series of parametric design studies on well-known buildings and building components have emerged [30,31,32]. In the field of architectural sustainability, parametric design is often combined with physical environment simulation technology to derive parametric design through environmental and climate parameters [33,34,35]. In the field of landscape design, under the influence of complexity theory, the characteristics of complexity and diversity and the method of parametric design have played an important role [36]. Traditional landscape design often relies on the subjective and empirical analysis and judgment of professionals. With the popularity of parametric design, many researchers began to conduct parametric design research on complex elements in landscape design [37]. The traditional design methods of Jiangnan gardens are often expressed by poetic words in classical literature, and there are no specific design parameters for designers to refer to, which makes it difficult for modern designers to design Jiangnan garden-style landscapes. In existing studies, researchers have conducted quantitative research on the natural and environmental characteristics of Chinese classical gardens [38]. However, the application of the parametric design method to Chinese classical gardens still needs to be explored, because it will not only help landscape architects to improve design efficiency, but also help to preserve the characteristics of classical gardens.

### 1.3. Problem Statement and Objectives

On the whole, although the method of road organization is complex, scholars have entered into quantitative exploration, and the corresponding indicators and path design rules have achieved certain results. However, considering the visual space perception of “moving and changing scenery” in classical gardens and the more challenging continuity design of traditional garden characteristics, this area of research still needs to be developed and studied. Artificial intelligence and parametric design are popular research topics at present. Parameterized and algorithmic intelligent generation of garden road system research has great exploration value for the present and future.

Tracing it back to the source, one of the keys to solving the problem also happens to include capturing the characteristics of classical gardens, perceiving the essence of garden paths, and applying traditional wisdom to modern landscape design. To this end, by studying the distribution characteristics of roads in traditional gardens (corresponding to the Hamiltonian cycle) and their tortuosity (curvature of road), the characteristic data of traditional gardens and roads represented by Jiangnan gardens were collected. At the same time, the obtained data were transferred to the parameterized platform, and a method of intelligent generation was used for calculations. Finally, the “wisdom of design” was realized through the application of drawing modern landscape designs through genetic algorithms. In this paper, the researchers explored the following four questions:(1)What are the distribution characteristics of traditional garden roads in Jiangnan classical gardens that can help implement intelligent garden road design?(2)How can we combine the collected road curvature, angle, and field of view data into the algorithm model?(3)How does the genetic algorithm assist in the intelligent design of garden roads?(4)How does this method work when it is used to build courtyards, community parks, and urban parks?

## 2. Materials and Methods

### 2.1. Method Process

This paper proposes a method, based on parametric design, to realize the generation of outdoor paths from the site outline and basic elements of the Jiangnan classical garden style. Additionally, it provides applications for the renovation, restoration, upgrading, design, and other aspects of garden projects focusing on the traditional Chinese style (Figure 1). Specifically, these findings can be used in courtyards, community parks, urban parks, and other spaces, and have important practical value for the unification of regional styles in garden design and the protection of garden features.

The method of this study consists of four aspects: feature acquisition, parametric translation, algorithm optimization, and model application. The details are as follows (Figure 1):(1)Feature extraction: First, it is necessary to clarify the specifications and characteristics of Jiangnan classical gardens. In this study, based on previous statistics, the area of Jiangnan classical gardens is taken to be larger than 3000 square meters, and the grid is divided into portions of 10.24 square meters (3.2 m × 3.2 m) as area units (according to the traditionally constructed measurements) (In the construction measurements of the Ming and Qing Dynasties, one *chi* = 32 cm and one *zhang* = 3.2 m). Secondly, the roads in Jiangnan classical gardens can be divided into three types: roads, corridors, and bridges. According to the records from The Craft of Gardens, the curvatures of different types of roads have different rules [39]. All roads in the Jiangnan classical gardens are counted, and their curvature is converted into radians to be utilized as one of the constraints and generation rules of the parametric design. Finally, the study found that the roads in Jiangnan classical gardens have various uses, including daily life functions, sightseeing functions, rest functions, and gathering functions, and these functions are mainly related to the spatial scale and field of view of the garden. The view of the bridge is the widest, the view of the road is the second widest, and the view of the corridor is the smallest.

(2)Parametric transfer: The parametric design uses Rhino’s Grasshopper as the software platform. Grasshopper is one of the most commonly used parametric software platforms. It can be utilized for visual programming, and it is widely used in the field of design and analysis [40]. According to the characteristics of classical gardens and the needs of modern landscape design applications, the application’s targets are positioned as three types: courtyards (3000 to 10,000 m^2^), community parks (10,000 to 20,000 m^2^), and urban parks (20,000 to 25,000 m^2^). From these three types of applications, the base elements for parametric design are extracted, such as site boundaries, points of interest, existing buildings, and lakes. The translation process is mainly divided into road generation, landscape area generation, landscape gallery generation, and landscape trail generation.①In road generation, the shortest path is generated according to the site boundary and points of interest. The path divides the site into a network according to the grid cell (10.24 m^2^/cell) of the Jiangnan classical garden, uses the Hamiltonian cycle [41] to generate a new path, optimizes the path according to the known curvature, and, finally, further adjusts the design of the obtained road according to the viewshed analysis.②In the generation of the landscape area, firstly, the core landscape area is generated with the inherent buildings and the above-generated roads as constraints, and, secondly, the view area is analyzed for the path of the road to obtain the minimum and maximum view areas. The two fields of view are superimposed, and finally, a semi-open landscape area, an open landscape area, and a private landscape area are obtained.③In the generation of the landscape gallery, according to the generated road and landscape area, the building and lake elements are imported at the same time. The road segment where the lake and the road intersect will generate a bridge; then, the viewshed analysis is performed around the lake and the road, and the maximum boundary of the overlapped viewshed area will generate a corridor. In addition, the corridors of Jiangnan classical gardens are usually not a closed loop (only about half), and the paths of some corridors can be deleted according to the actual situation of the site.④In the generation of landscape trails, there are usually many landscape features that adorn the landscape space in Jiangnan classical gardens, such as stacked rocks, rockeries, precious plants, etc. [42] The number of these landscape features is determined according to the financial resources of the garden owner. Landscape features can be used as necessary points in landscape trails, and landscape trails can be generated with the slime mold algorithm. The slime mold algorithm is one of the optimization algorithms which simulates the spreading and foraging behavior of slime mold, and is also commonly used in the field of simulating biological behavior to find the optimal path [43]. By simulating the slime mold algorithm three times, the simulation results can be superimposed on each other, and finally, the landscape trail is generated.

(3)Algorithm optimization. In order to further make the results generated by the parametric design conform to the characteristics of Jiangnan classical gardens, the roads and corridors from the above-generated results are optimized via a genetic algorithm. The genetic algorithm is one of the algorithms for solving unconstrained and constrained nonlinear optimization, one which can effectively solve the problem of global optimization [44].①In the optimization of the roads, the genetic algorithm is used to find the combination of the maximum field of view and the minimum path length based on the radian of the road, with Jiangnan classical gardens as the optimization result.②In the optimization of the corridors, the genetic algorithm is used to find the combination of the maximum field of view and the maximum path length based on the radian of the road, with the Jiangnan classical garden as the optimization result.③An artificial judgment is made on all the above-generated results, as the numerical optimal result of the parametric design may not be the optimal design result. The principles of judgment: Did the effect of the visual field achieve the expected goal? Is the overall layout reasonable? Can the final design meet the needs of the designer, party A, and the design specification, such as the road network density required by the design specification? If the results do not meet the above principles, they can be adjusted again.(4)Model application: According to the above process, the model can be applied to courtyards (3000 to 10,000 m^2^), community parks (10,000 to 20,000 m^2^), and urban parks (20,000 to 25,000 m^2^). The differences between the three are not only reflected in the scale of the area, but also in their inherent buildings, water bodies, points of interest, and must-pass points. The model can be well adapted to the differences between these elements and has practical value in both new landscape design and old landscape renovation.

### 2.2. Analysis of Path Characteristics

#### 2.2.1. Road Types and Characteristics

The Ming and Qing Dynasties were the mature period in the development of Jiangnan classical gardens in China, and when the unique gardening art of Chinese classical gardens was most highly developed. The objects of this study are nine typical classical gardens in the Jiangnan region of China: Humble Administrator’s Garden (The Humble Administrator’s Garden is a UNESCO World Heritage Site and one of the most famous of the gardens of Suzhou.), Lingering Garden (Lingering Garden is a renowned classical Chinese garden, dating back to 1593. It is located at 338 Liuyuan Rd.,. Suzhou, Jiangsu Province, China. Since 1997 it has been recognized with seven other classical gardens of Suzhou as a UNESCO World Heritage Site. The garden also contains two UNESCO Intangible World Heritage Arts: Pingtan and Guqin music.), Canglang Pavilion (The Canglang Pavilion was built in 1044 CE by the Song dynasty poet Su Shunqin (1008–1048) on the site of a pre-existing imperial flower garden c. 960 CE. It is the oldest of the UNESCO gardens in Suzhou, keeping its original Song dynasty layout.), Master of the Nets Garden (Master of the Nets Garden, previously called Ten Thousand Volume Hall, was first constructed in 1140 by Shi Zhengzhi. It is recognized with other classical Suzhou gardens as a UNESCO World Heritage Site.), Garden of Harmony, (The Garden of Harmony, otherwise called Yuyuan, was built in the Qing Dynasty by Gu Wenbin, an official from the Qing Dynasty.) Lion Forest Garden (The Lion Grove Garden is a garden located at 23 Yuanlin Road in the Gusu District (formerly Pingjiang District), Suzhou, Jiangsu, China. The garden is famous for the large and labyrinthine grotto of Taihu rocks at its center.), Retreat and Reflection Garden (The Retreat and Reflection Garden is a notable classical garden in China.), Couple’s Retreat Garden, and Jichang Garden is a famed Chinese classical garden, and it was claimed as a nationally protected location of historical and cultural relics on 13 January 1988.), which were selected for special analysis of their internal roads (Figure 2).

“The scenery is different when you walk, and the winding path leading to the secluded” is a typical feature of the paths of Jiangnan classical gardens. Since ancient times, gardening in the Jiangnan classical gardens has paid attention to the relationship between architecture and mountains, rivers, and trees, and pursued the creation of natural beauty without artificial traces in the garden landscape. As an important connection between building and landscape, the manner of its reflecting the beauty of nature over the course of a path is particularly important for its layout design.

There are various types of roads in Jiangnan classical gardens, consisting of garden paths, corridors, and bridges. Among them, garden paths are the main road-type in landscape gardens. The garden path is located within a landscape of mountains, stones, flowers, and trees, and it is provided as a way for the owner of the garden to enjoy the landscape. The corridor is a unique type of road in Jiangnan classical gardens. It is usually located between the main buildings, connecting them together. The climate in the Jiangnan classical gardens is humid and rainy, and the corridor, as a roofed road, acts as a shelter from wind and rain, ensuring accessibility between buildings that mainly have residential functions. At the same time, the zigzagging of corridors within the landscape can also provide a good viewing experience. The bridge is the water extension in Jiangnan classical gardens, such as can be seen in the roads of the Humble Administrator’s Garden. There are lakes and water systems to the south of the Yangtze River, and the water area in the garden is relatively large. While the bridge is used for water transportation, it becomes a unique landscape presentation in the garden, together with the water (Figure 3).

#### 2.2.2. Path Function and Distribution Characteristics (Spatial Sequence)

Jiangnan classical gardens are built in the city; thus, one question is: how to create a natural landscape suitable for people’s lives in the cramped urban space? This is the same problem found in modern urban landscape construction. In order to create a rich and varied landscape-effect in a limited space, the roads of Jiangnan classical gardens have unique layout characteristics.

The first layout is the “space loop”, in which the road provides accessibility with regard to the residential function. Jiangnan classical gardens usually have a loop through each functional building, connecting each functional area in a series. This loop is connected, in turn, by garden paths or corridors, forming the main road in the garden. For example, the road system in the Lingering Garden seems to be very complex, but from the central part of the paths, it can clearly be seen that the garden roads are arranged in a circular sequence, and the corridors and garden paths each form a spatial loop with the buildings (Figure 4).

The second is the “winding and changing” layout, which mainly provides a tourist function. The layout of the corridor is characterized by winding and twisting, and adopts the method of changing roads. Various types of buildings, such as corridors, pavilions, and waterside pavilions, are arranged around the water’s surface along the perimeter of the garden, forming a visual experience that is centripetal and cohesive. At the same time, the twists and turns of the corridor are added to achieve the effect of changing and continuous scenery. The moving back and forth on the path is tortuous, which invisibly increases the length of the journey and the tour time, and psychologically extends the original sense of space. Walking in the winding and changing corridors and watching the garden scenery on both sides is like looking at a magnificent scroll of Chinese landscapes.

The third is the “degree of twists and turns” layout, which plays a role in enriching pedestrians’ viewing experiences. Appropriate twists and pauses can adjust the walking rhythm between movements, stops, and turns in the process of walking. The corridors and bridges are usually arranged in a “zigzag” shape, although the zigzag angles are usually not rigid right angles. The junction between the corridor and the wall is transformed into a pavilion or landscape of mountains, stones, flowers, and trees that functions as an embellishment and as a node for road rests. For example, the “Moon Come With Breeze Pavilion” in the Master of the Nets Garden not only enriches the experience of the road, but it also breaks the sense of occlusion caused by the flat and rigid walls of the original wall.

### 2.3. Garden Road Curvature and Viewshed Analysis

According to the characteristics of Jiangnan classical garden roads described above, nine typical Jiangnan classical gardens were selected as the research objects for this paper. First, using the two aspects of garden road tortuosity and viewing experience as the starting point, quantitative statistics were collected on the garden roads’ tortuosity and their viewing fields in order to determine the inner law of the garden roads and how they form the feeling of the “winding path leading to secluded, moving with different scenery”. In addition, the statistical results were incorporated into the intelligent generation of road parameters in this paper to form a landscape road system with the characteristics of classical gardens.

#### 2.3.1. Road Analysis Methods

The curvature and angle of the corner directly affect the shape and walking experience of the path. If the characteristic of the garden road is the “winding path leads to secluded”, then the turning curvature and angle of the road are the keys. Therefore, this paper calculated the curvature of garden roads according to three types of garden roads (for the raw statistical data, refer to Table A1 and the summary of nine garden roads in Appendix B): the garden path (refer to Table A2, Table A3, Table A4, Table A5, Table A6, Table A7, Table A8, Table A9 and Table A10 in Appendix B for the raw statistical data), the corridor (refer to Table A11, Table A12, Table A13, Table A14, Table A15, Table A16, Table A17, Table A18 and Table A19 in Appendix B for the raw statistical data), and the bridge (refer to Table A20, Table A21, Table A22, Table A23, Table A24, Table A25, Table A26, Table A27 and Table A28 in Appendix B for the raw statistical data). This method of analyzing the curvature of the road was used to obtain the coordinate values of all turning points from the road’s centerline, and to calculate the curvature and angle of the corners formed by a group of three adjacent coordinated points, thereby obtaining the curvature and turning angle values of the garden road. In addition, visual observation points were set at each corner, and the viewing area of each observation point in the garden was calculated and counted.

#### 2.3.2. Road Curvature Analysis

First, we determined the statistics of curvature, which mainly express the “curvature degree” of the road. By calculating the curvature statistics of the median lines of each road in nine typical gardens, it was found that: the average curvature of garden paths was 0.34, the average curvature of corridors was 0.28, and the average curvature of bridges was 0.20. As far as the overall curvature value is concerned, it can be seen from Figure 5 that the curvature of the garden path had the largest fluctuation range, and its curvature range was 0~0.81 (Figure 5, garden path). The curvature of the corridor ranged from 0 to 0.65 (Figure 5, corridor). The curvature of the bridge ranged from 0 to 0.41 (Figure 5, bridge). It can be seen from the data chart that the tortuous degree of the garden path was the largest, and there were also more bending changes; the tortuous degree of the corridor was relatively stable, and the tortuous degree of the bridge was the smallest, given that there were fewer bending changes.

#### 2.3.3. Road Angle Analysis

The turning angle corresponding to the curvature can more intuitively show the “turning degree” of the garden road system. From the turning angle of the garden road system (Figure 6), it was determined that the turning angle of the garden path was 58.36~180°, the turning angle of the corridor was 75.37~180°, and the turning angle of the bridge was 118.97~175.10°. The range of folding angles of the garden path was large, with the exception of a few relatively sharp angles at 58.36° and 72.13°; most folding angles averaged around 129.40° (Figure 6, garden path). The range of the folding angles of the corridor obeys a certain law, and it can be seen that the frequency of about 85~90° appears at the starting point and the ending point. This shows that almost every section of the corridor will have a small section that follows the edge of the building and presents a phenomenon similar to a right angle. Most of the angles averaged around 130.26° (Figure 6, corridor). The bending angle range of the bridges was relatively small, and the turning angles of all bridges were obtuse, with an average of 150.84° (Figure 6, bridge).

#### 2.3.4. Degree Analysis of Garden Road System Viewsheds

In order to further study the visual perception of the scenery in the process of road walking, we must consider how to reflect the garden road system characteristics of the “moving and changing scenery”. This paper quantified “movement” as the length of the movement during walking, and “view” was quantified as the viewing area at the turning point of the path. Further statistical research was conducted on the turning length and viewing area of each turning point in the process of road-walking (Figure 7).

Based on the previous garden road system angle statistics, the researchers analyzed the relationship between the average angle of the overall garden road system and the average viewing area of the nine Jiangnan classical gardens. In Figure 8, it can be clearly seen that the average turning angles of the three road types in the park are relatively close, and the fluctuation trend is small (Figure 8, bar chart). Relatively speaking, the average viewshed area value of each park road varies greatly (Figure 8, area map). Interestingly, the Humble Administrator’s Garden had a high landscape visibility in terms of the total average viewing area, with an average range of about 3797.09 to 5461.93 square meters, while the average viewing area of the rest of the gardens ranged from 965.87 to 1874.78 square meters.

From this analysis, it can be seen that, under the condition of similar angles, the total viewing area of the Humble Administrator’s Garden is much larger than that of the other gardens. Therefore, this study took the Humble Administrator’s Garden as the example of typical paths for the aspect of “moving and changing scenery”, and chose the Retreat and Reflection Garden as the representative garden to carry out a comparative study on the field of view area and road length. Further comparisons were made between the turning unilateral lengths (edge lengths) of various types of garden paths in the Humble Administrator’s Garden and the Retreat and Reflection Garden and the viewing area. Thus, the road length and viewing area charts were formed.

From the perspective of the turning length of the garden path, the average total length of the garden road in the Humble Administrator’s Garden was 31.21 m, the turning unilateral length was between 0.74 and 22.58 m, and the average viewing area was 4313.55 square meters. The average total length of the corridor was 31.15 m, and its average viewing area was 4017.45 square meters. However, unlike the value of the garden path, the length of the corridor was between 2.26 and 13.60 m. The average total length of the bridge was 11.09 m, the unilateral length of the turning point varied between 0.48 and 4.58 m, and the average viewing area was 6121.33 square meters (Figure 9).

The average total length of the garden path in the Retreat and Reflection Garden was 19.57 m, the turning unilateral length was between 0.79 and 11.46 m, and the average viewing area was 1307.02 square meters. The average total length of the corridor was 28.37 m, the unilateral length of the turning point varied from 0.68 m to 20.94 m, and its average viewing area was 1110.01 square meters. The length of the corridor was about 4~5 m. The average total length of the bridge was 7.38 m, the unilateral length of the turning point varied between 2.15 and 2.91 m, and the average viewing area was 1864.34 square meters (Figure 10).

To summarize, we can represent the relationship between the road turning length and the horizon as follows:Garden paths: walk less, see more, and change more. The total walking length of the garden path is short, the turning length changes more, and the visual field changes abundantly.Corridor: walk a lot, see a lot, and change a lot. The total walking length of the corridor is long, and the transitions and sights have a certain sense of continuity and rhythm.Bridge: walk less, see more, but change less. The total walking length of the bridge is short, the turning length is uniform, and the field of vision is the widest.

Through the above statistical analysis of curvature, angle, and viewing area, this study obtained the characteristics and angle range values of the road turning changes. The angle range value, turning length, and view field law of these garden paths, corridors, and bridges are incorporated into the parametric generation later in this paper as an important basis for affecting the generation of garden road system corners.

## 3. Parametric Construction and Generation Process of a Garden Road System

### 3.1. Algorithms and Generation Process

This paper proposes a method of generating paths for Jiangnan classical gardens based on parametric design. The method is based on the analysis and extraction of path features of nine typical classical gardens, such as path type, function, distribution characteristics, and path radian. At the same time, various methods such as the Hamiltonian cycle, viewshed analysis, and slime mold algorithm were used to derive the path generation of the garden, and the genetic algorithm was used to obtain the globally optimal path design.

The landscape characteristics of Jiangnan classical gardens can be divided into four dimensions, namely, coherence, naturalness, complexity, and historicity [45].

Coherence is defined as the coherence of space, that is, “space loop”.Naturality emphasizes the interaction and connection between people and natural elements (lakes, plants).Complexity enriches the form of landscape elements.Historicity emphasizes the traditional gardening rules of Jiangnan classical gardens.The dimensions of the above landscape features can be translated into parameterized parameters: roads, landscape areas, corridors, bridges, trails, radians, etc. (Figure 11).

**Figure 11 ijerph-20-03158-f011:**
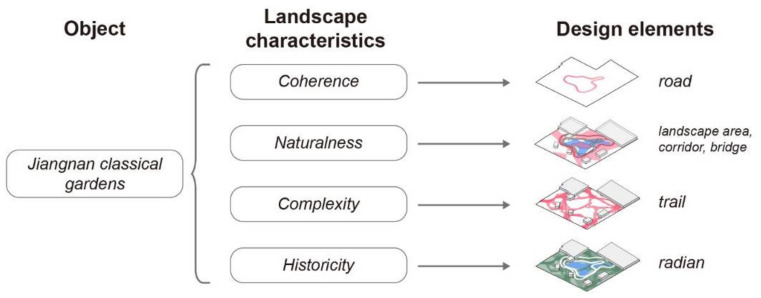
The dimensions of landscape characteristics of Jiangnan classical gardens.

(1) For the embodiment of the coherence dimension of Jiangnan classical gardens, the method of the Hamiltonian cycle was adopted in the parametric design.

Specifically:Set up points of interest in the site, which can be pavilions, platforms, and buildings in Jiangnan classical gardens (Figure 12(1)).Divide the site into several 3.2 m × 3.2 m grids to generate an undirected graph (Figure 12(2)).Then, use an algorithm to concatenate these points of interest along the grid lines (Figure 12(3)). The traditional series connection method will cause some roads to overlap and have too many long and straight lines, which is not in line with the characteristics of Jiangnan classical gardens.Our method is based on the Hamiltonian cycle and adds new rules: roads cannot overlap and polylines are maximized (Figure 12(4)).This method can automatically generate more complex and coherent road loops for the prototype of the road system according to the set parameters.

(2) To reflect the natural dimension of the Jiangnan classical gardens, the visual area is used as the main judgment basis in the parametric design.

Specifically:Place sight-blocking objects in the site, which can be rockeries, rocks, and plants (Figure 12(5)).Analyze the field of view of the site and calculate the position and area of the largest field of view in the road as an open space (Figure 12(6)).Further, carry out the analysis of the field of view, and calculate the position and area of the minimum field of view in the road as a private space (Figure 12(7)).Superimpose the above analysis results on the visual field. The overlapping part is set as semi-open space, and the rest is open space and private space (Figure 12(8)).

Different space attributes can be set with different landscapes to help reasonably plan the activity space. For example, set up more event facilities (porches and bridges) in open spaces. In private spaces, a richer natural landscape can be set, and the semi-open space is a compromise between man-made objects and natural objects.

(3) To reflect the complexity dimension of Jiangnan classical gardens, the slime mold algorithm (SMA) method was used in the parametric design. The slime mold algorithm simulates the behavior and morphological changes of Physarum polycephalum during the foraging process. It can ensure the diversification and efficiency of the path, achieving balance between the two [46].

Specifically:Set the starting point and food locations in the venue (Figure 12(9)). The starting point is the entrance to the site. The trail acts as an extension of the main road and is therefore set on the main road.Start the slime mold so that the slime mold grows from the starting point and forms a path through processes such as covering food locations, extending outward, and path oscillation (Figure 12(10)).After a period of time, the slime mold will connect all the food locations in the field, and the changing shape will gradually become stable (Figure 12(11)).Stop the growth of the slime mold (Figure 12(12)).

The growth trajectory of slime molds can be used as a reference for the design of garden trails. Each activation of the slime mold yields different results, allowing for different trail designs to be derived. This bionic design derivation process reflects the complex dimension of Jiangnan classical gardens.

### 3.2. Generation Process

The process of scheme generation was implemented via the Grasshopper visual programming application in Rhinoceros 3D modeling software to set various parameters and formulate generation rules through programming in order to realize the final generation of the scheme (refer to Appendix A: Grasshopper environment configuration).

(1) Site infrastructure construction (Figure 13(1)):

Set the basic conditions of the site, delineate the boundaries of the site, and put the buildings and water bodies that need to be preserved in the current situation into the model (Figure 13(1-1)).

Establish the basic grid (the size of each grid is 3.2 m × 3.2 m) and the generated range (Figure 13(1-2)).

Generate the shortest path by means of the Hamiltonian cycle (Figure 13(1-3)).

Classify the polylines and long straight lines of the shortest path, make the polylines generate smooth curves, and adjust the curvature of the curves (Figure 13(1-4)).

Finally, integrate straight lines and curves as the prototype of the main garden road.

(2) Main garden road generation and optimization (Figure 13(2)):In Jiangnan classical gardens, the garden road will be adjusted according to the special features in the garden so that the road is closer or farther away from the features, allowing you to set interference points to fine-tune the garden road (Figure 13(2-1)).According to the width of the main road, make the line segment generate the road segment (Figure 13(2-2)).Perform a viewshed analysis on the generated main road, recording the maximum viewpoint, the minimum viewpoint, and the viewport area as the parameters of the subsequent design (Figure 13(2-3)).Use the genetic algorithm to optimize the main garden road (Figure 13(2-4)).

The curvature value of the main road is controlled, and parameters such as the viewing area and path length are used as target values so that the genetic algorithm can obtain the final curvature value under the condition that the viewing area is the largest and the path length is the smallest through continuous iteration. At the same time, this value also needs to meet the curvature range of Jiangnan classical gardens.

(3) Corridor generation and optimization (Figure 13(3)):Derive the scope of the center of the visual field from the path of the main garden road as the area for generating the core landscape (Figure 13(3-1)).According to the distance between the corridor and the road in the Jiangnan classical garden, the path offset by the main garden road generates the corridor path (Figure 13(3-2)).Perform viewshed analysis on the path of the corridor, recording the maximum viewpoint, the minimum viewpoint, and the view area as the parameters of subsequent design (Figure 13(3-3)).Use the genetic algorithm to optimize the corridor’s path. Control the curvature value of the corridor path and access the viewpoint area and path length parameters as the target values (Figure 13(3-4)).

Let the genetic algorithm obtain the final curvature value under the conditions of the largest viewing area and the largest path length through continuous iteration. At the same time, the value also needs to meet the curvature range of Jiangnan classical gardens.

(4) Generation and local optimization of garden paths and bridges (Figure 13(4)):Set the site entrance as the starting point, and the buildings and main road nodes in the site as food locations. Then, activate the slime mold algorithm so that the slime mold covers all food locations in the field. The spawning trajectories of slime molds act as a scheme for garden paths. The slime mold algorithm can be restarted continuously to generate different scenarios, and the appropriate one can be chosen as the final garden path (Figure 13(4-1)).Set the road across the water as a bridge (Figure 13(4-2)).Partial adjustments are made to the path of the corridor, and the corridor can be increased or decreased according to the project cost of the garden road and the needs of landscape zoning. The corridors in the Jiangnan classical gardens are usually not a complete path, and about half of the closed-loop path can be cut (Figure 13(4-3)).Perform visual field analysis on the path of the bridge to see if the visual field of the bridge can cover important landscape areas to meet the needs of people viewing the scenery from the bridge (Figure 13(4-4)).

### 3.3. Effectiveness Assessment

After optimizing the generation process through the above parameterization and algorithm, the generated road can be compared with the original road for evaluation. Taking the Retreat and Reflection Garden as an example, it can be seen intuitively from Figure 14 that the layout of the newly generated road is very similar to the layout of the original path. At the same time, the calculation results of the best viewpoints for the three types of roads, namely, the garden path, corridor, and bridge, are also very close, especially as the best viewpoints of the garden path and corridor are almost the same. The newly generated bridge has an additional viewpoint compared to the original, and the result of the viewshed calculation is that the new bridge has a higher viewshed degree (Figure 14).

From the data chart, it can also be seen that the effect of the parameterization and optimization algorithms not only continues the fluctuation trend and characteristics of the curvature and turning angle of the original path, but also increases the viewing area during the path-walking process. Comparing the curvature and angle values of the original road and the parametrically generated road, it was found that the angle value range of the original garden path was 90.71~162.57°. The parameterized garden path angle interval value is 94.97~171.34°. The original corridor path angle range was 90.00~168.29°, and the parameterized corridor angle range is 109.97~168.02°. The original bridge angle range was 136.95~142.49°. There are two bridges generated by parameterization, and the angle range values are 124.96~141.97° and 144.18~149.97°. Similarly, the road curvature of the parametrically generated road is very similar to the original path value (Figure 15 and Figure 16).

Evaluated from the viewshed aspect, the parametrically generated road has a larger viewshed visible area than the original road. The maximum areas of the original three types of roads were: garden path, 1704.42 m^2^; corridor, 1528.73 m^2^; and bridge, 1953.55 m^2^ (Figure 17). The maximum areas of the three types of road views generated by parameterization are: garden path, 1655.07 m^2^; corridor, 1580.87 m^2^; and bridge, 1655.20 m^2^ (Figure 18). After comparing the floor plan with the data, the research was shown to have obtained relatively intuitive and rational evaluation results. The results from the road curvature, angle, and field of view are obvious, which shows that the method of this study is feasible.

## 4. Discussion and Results: Application Effectiveness

In order to reflect the feasibility and application value of this research in modern applications, this research effort selected three practical projects and further applied the evaluated parameterized generation logic and optimization algorithm to them. They include 3000~10,000 square meters of courtyard green space (equivalent to the area of the Retreat and Reflection Garden), 10,000~20,000 square meters of community parks (equivalent to the area of the Canglang Pavilion), and 20,000–25,000 square meters of urban parks (equivalent to the area of the Humble Administrator’s Garden).

Figure 19 shows the application effect of this study on courtyard green space (5307.29 m^2^), a community park (16,859.58 m^2^), and an urban park (22,598.35 m^2^). The site status was imported into Rhino, including red lines, reserved buildings, reserved landscapes, and water. According to the above parameterization generation logic and optimization process, the road system with the characteristics of classical gardens can be automatically generated (Figure 19).

From the perspective of the application effect, it was successful. The road-system plan generated by the algorithm according to the current project inherited the characteristics of classical garden roads. Of the three applications, the road generation method obtained in this study has been applied to practical projects, and the construction of the urban park project in Figure 19c has started. The urban green space is part of Shanghai Pudong’s urban greening system. The park is located within Hengmian, Pudong, Shanghai. It is positioned as a park with Jiangnan garden characteristics, which will be completed and opened in early 2023. Therefore, our research results can be applied in practical projects. This not only verifies that the method of this study is feasible, but also that it has practical application value (Figure 20).

Nonetheless, our methods leave much to be desired. For example, not all the results obtained by the algorithm can be used directly. If the road is too complicated or broken, the designer needs to fine-tune it according to its actual situation. The actual uses of these findings face some limitations at present, and there are many areas to be improved in the future.

## 5. Conclusions

This paper took the garden road as a research object and proposed a method of generating paths for classical gardens based on parametric design. Through the practical application of research and design projects, the following conclusions were obtained:(1)A parametric platform with Grasshopper as the core can effectively program the design of Jiangnan classical garden roads and reflect the relevant spatial characteristics through data input.(2)After the programming of the Grasshopper parameterized platform has been completed, the global road network system can be further adjusted through the genetic algorithm to make the landscape more abundant.(3)The slime mold algorithm has bionic characteristics, which has been reflected in the research of Jiangnan classical gardens, since ancient Chinese gardens have the characteristic of being integrated into the natural environment; however, if modern garden design utilizes the form of geometric cutting, this algorithm is not applicable.(4)Judging from the effects of the projects constructed using landscape design practices that the researchers are developing at present, the method established in this study is effective.

The development of artificial intelligence has improved people’s efficiency and lessened the hardship of boring and repetitive work. However, it is difficult for computer programs to intelligently identify emotional factors in traditional culture. Chinese classical gardens carry the wisdom of the continuation of cultural thought, and modern design practitioners should consider the utilization of artificial intelligence from the perspective of sustainable design and the continuation of the historical context. This research not only identified the characteristics of landscape cultural heritage, but also developed an innovative, intelligent design tool. Urban green space roads are a part of urban systems. We hope that this intelligent generation method can be applied to more urban landscape spaces in the future.

## Figures and Tables

**Figure 1 ijerph-20-03158-f001:**
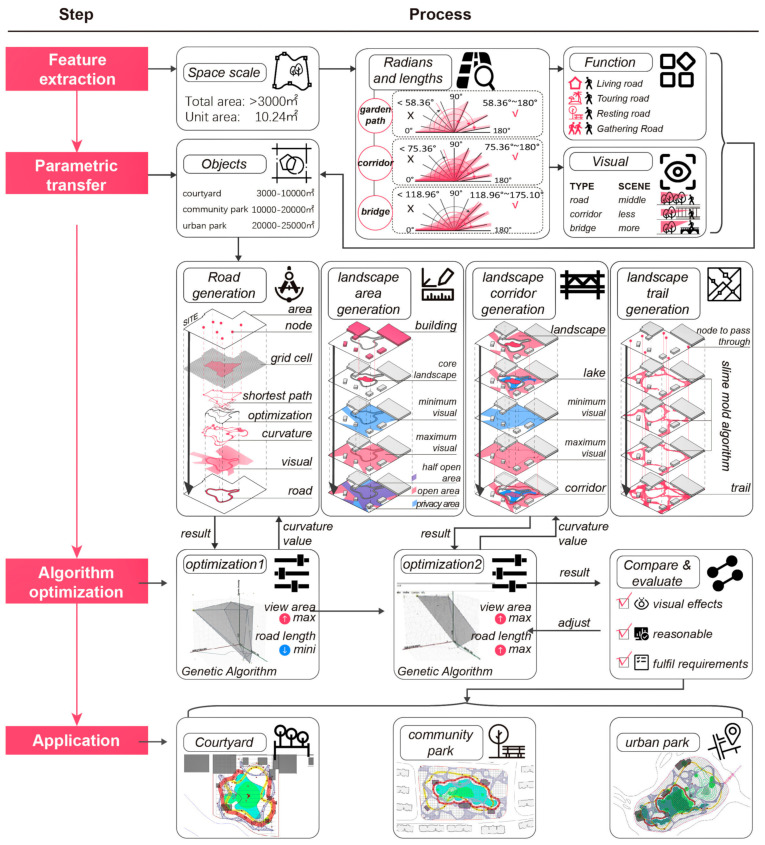
Method flow diagram.

**Figure 2 ijerph-20-03158-f002:**
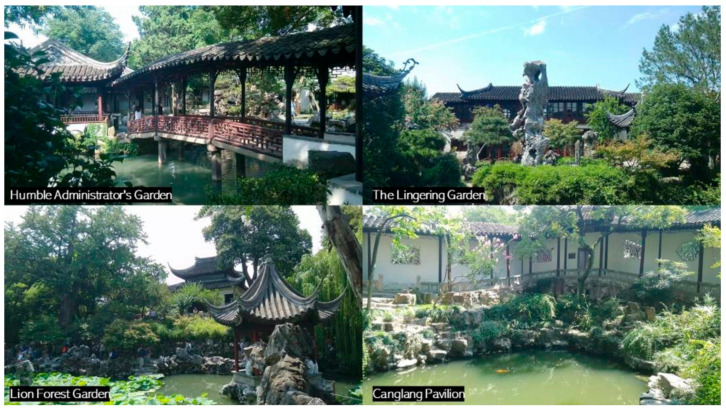
Typical representative images of Jiangnan gardens. (Image source: photographed and annotated by the author).

**Figure 3 ijerph-20-03158-f003:**
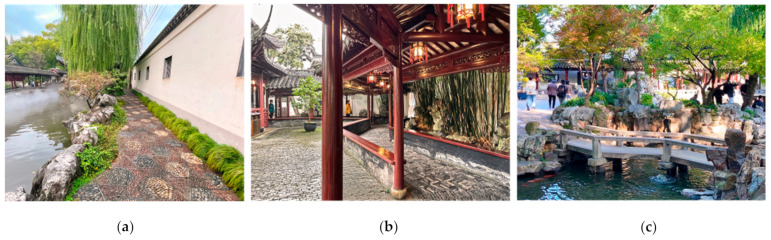
Types of roads in Jiangnan classical gardens: (**a**) garden path; (**b**) corridor; and (**c**) bridge. (Image source: photographed by the author.)

**Figure 4 ijerph-20-03158-f004:**
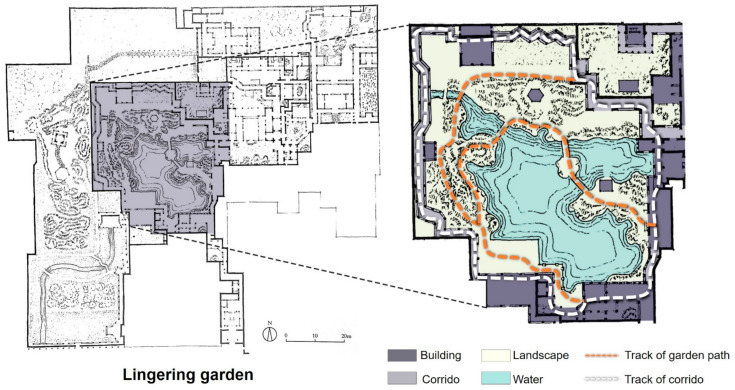
Floor plan of the central part of the Lingering Garden.

**Figure 5 ijerph-20-03158-f005:**
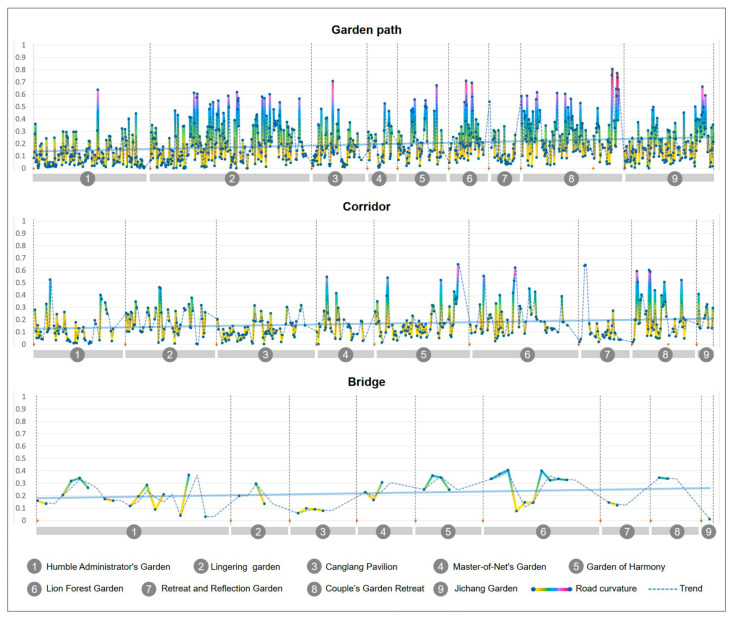
Road curvature analysis map of nine typical gardens.

**Figure 6 ijerph-20-03158-f006:**
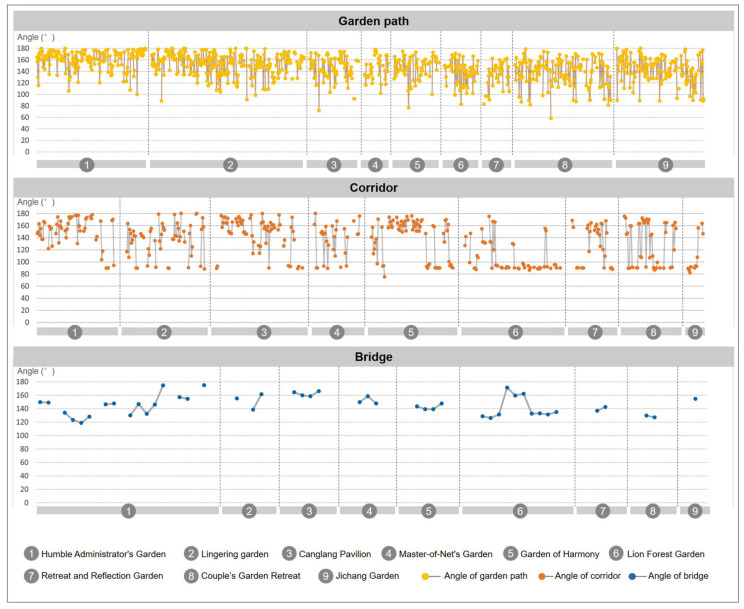
Nine typical garden road system angle analysis diagrams.

**Figure 7 ijerph-20-03158-f007:**
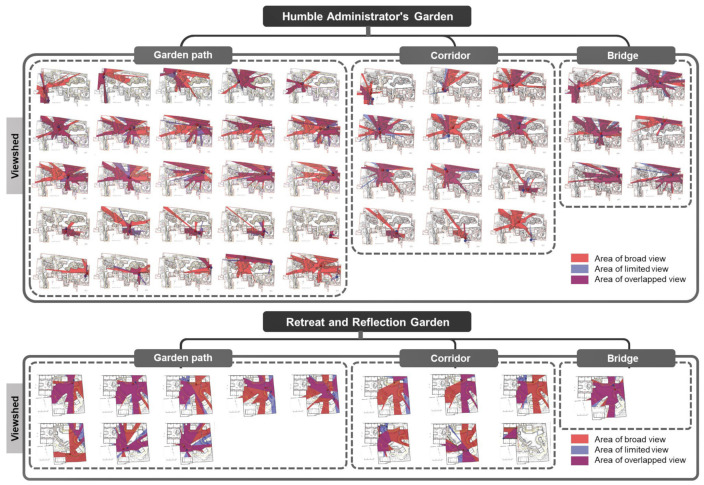
Road viewshed area analysis.

**Figure 8 ijerph-20-03158-f008:**
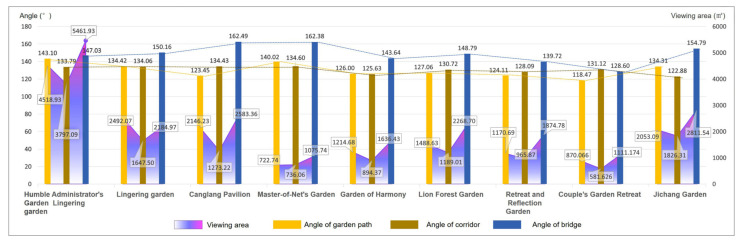
Average angle and average viewing area of road systems in nine Jiangnan classical gardens.

**Figure 9 ijerph-20-03158-f009:**
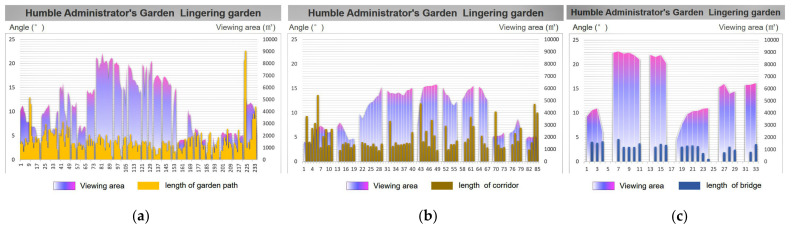
The Humble Administrator’s Garden: overlay map of unilateral length and viewshed area of road turns. (**a**) The garden path length and viewing area; (**b**) the corridor length and viewing area; and (**c**) the bridge length and viewing area.

**Figure 10 ijerph-20-03158-f010:**
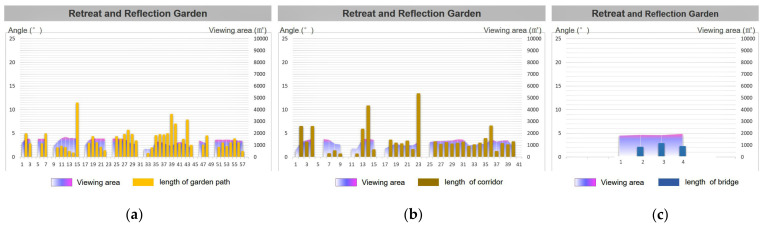
The Retreat and Reflection Garden: overlay map of unilateral length and viewshed area of garden road turns. (**a**) The garden path length and viewing area; (**b**) the corridor length and viewing area; and (**c**) the bridge length and viewing area.

**Figure 12 ijerph-20-03158-f012:**
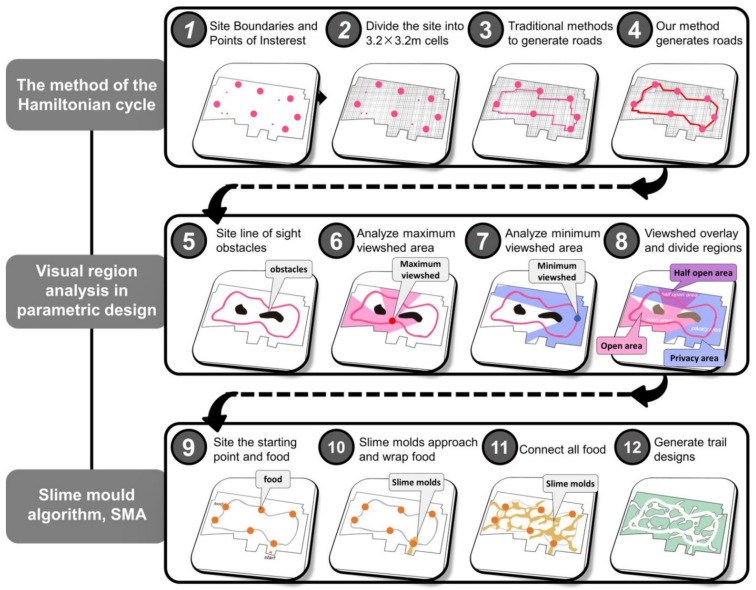
Design method and process (4) For the embodiment of the historical dimension of Jiangnan classical gardens, the genetic algorithm method is mainly used in parametric design. As seen in the prototype of the scheme generated in the process described above, the curvature and viewshed values obtained from the analysis are incorporated into the genetic algorithm to further optimize the path. At the same time, it is also necessary to control the density of the road network and appropriately increase or decrease the length of the path to meet the requirements of the design specifications. The curvature and field of view are obtained from the statistics of the existing Jiangnan classical gardens, and are also very important components of the Jiangnan classical gardens. The relevant laws have continued to exist throughout history, and the new scheme conforms to them, reflecting the historical dimension of the garden.

**Figure 13 ijerph-20-03158-f013:**
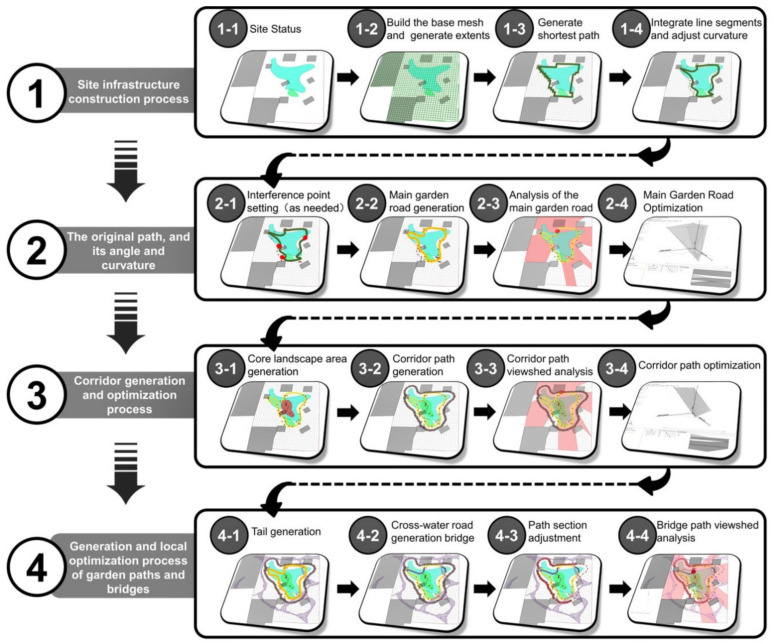
Parametric design steps and procedures.

**Figure 14 ijerph-20-03158-f014:**
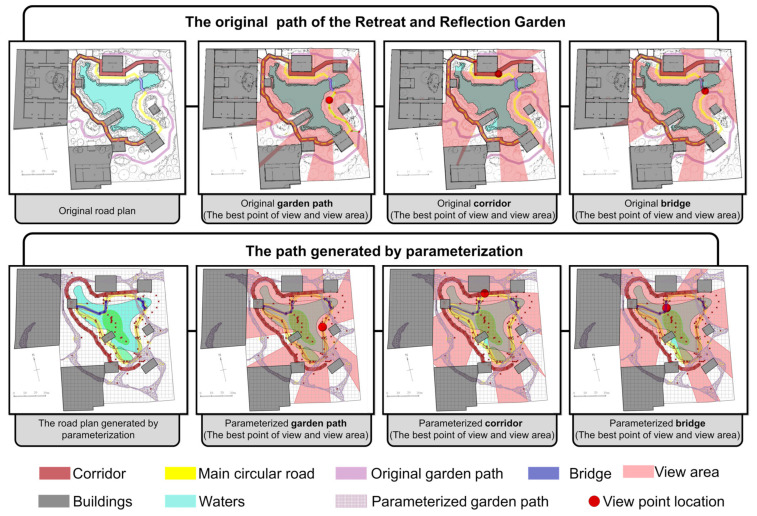
Comparison of original road and parametrically generated road in the Retreat and Reflection Garden.

**Figure 15 ijerph-20-03158-f015:**
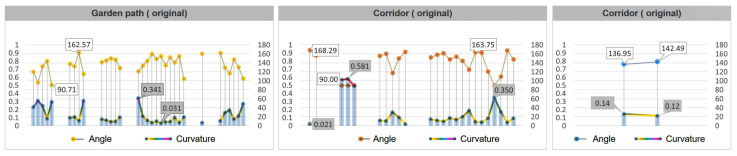
The original path and its angle and curvature.

**Figure 16 ijerph-20-03158-f016:**
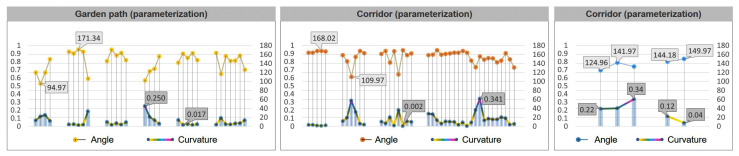
The path generated by parameterization and its angle and curvature.

**Figure 17 ijerph-20-03158-f017:**
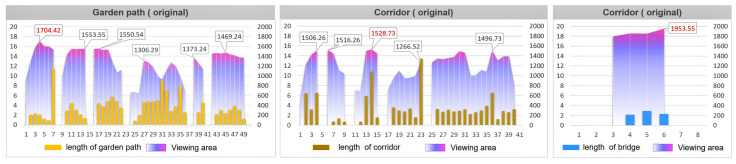
The superimposed comparison diagram of the length of one side of the road and the viewing area of the original garden path.

**Figure 18 ijerph-20-03158-f018:**
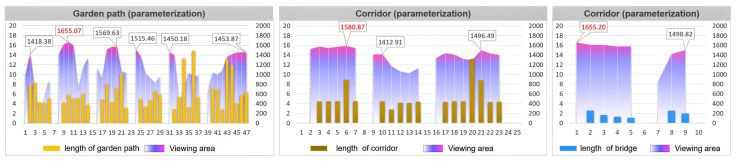
The comparison chart of the superposition of the length of one side of the road and the area of the viewshed generated by parameterization.

**Figure 19 ijerph-20-03158-f019:**
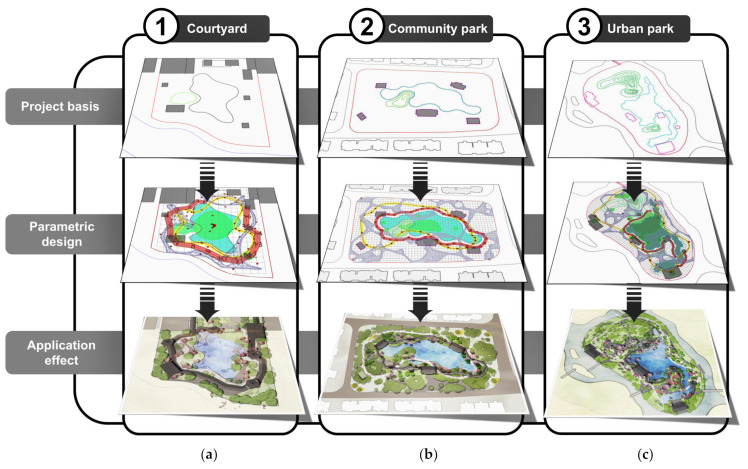
Parametric application effect: (**a**) application effect on courtyard green space; (**b**) application effect on community park; and (**c**) application effect on urban park.

**Figure 20 ijerph-20-03158-f020:**
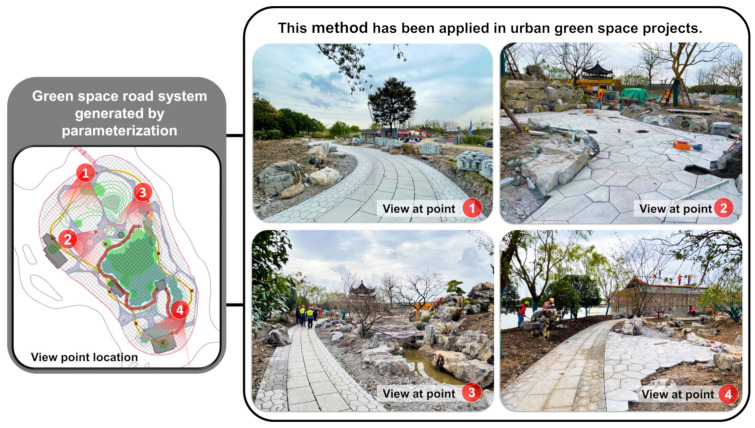
Application of intelligent road generation in urban green space. (Image source: photographed and annotated by the author.)

## Data Availability

The datasets used and analyzed during the current study are available from the corresponding author on reasonable request.

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
