# Peer review of "Intelligent Generation Method and Sustainable Application of Road Systems in Urban Green Spaces: Taking Jiangnan Gardens as an Example"

_ijerph, 2023, doi:10.3390/ijerph20043158_

Round 1

Reviewer 1 Report

The paper presents highly interesting research with an appropriate database and a well-described design method. However, there are minor issues that need improvement: 

1. The method section refers to the created design method, not the entire research process- first step - a literature review and background are needed. 

2. The above refers also to the introduction, where the theoretical background is not presented in the appropriate form. The background should involve for example methods for various greenery automation or other urban environment and landscape methods. Even if the Int J. Environ. Res. Public health or Sustainability there are several recent publications regarding the topic. Moreover, the concept of parametric design and generation of versions should be explained and placed in the context of recent research.  

The overall outcome of the research is of high quality and value. 

Author Response

Comments1: The paper presents highly interesting research with an appropriate database and a well-described design method. However, there are minor issues that need improvement:

The method section refers to the created design method, not the entire research process- first step - a literature review and background are needed.

Response:

Thank you for your comments. We have added the corresponding positions in the article. The revisions are available in the article. Especially in the part of the introduction, we added part 1.3 to make the logic of the research question clearer.

(line139-166)

Comments2: The above refers also to the introduction, where the theoretical background is not presented in the appropriate form. The background should involve for example methods for various greenery automation or other urban environment and landscape methods. Even if the Int J. Environ. Res. Public health or Sustainability there are several recent publications regarding the topic. Moreover, the concept of parametric design and generation of versions should be explained and placed in the context of recent research.

Response:

Thank you for pointing this out, which is very helpful for our revision.  Our previous thinking perspective may indeed be narrow, only focusing on the study of the garden road of Chinese traditional gardens.

Gardens have developed into modern open spaces (parks) that can be used by the public and are a kind of urban green space.  Therefore, we added a review of sustainable urban green space design methods and related references of parametrized application research in 1.2 Literature review.

(line63-81;line114-166)

Reviewer 2 Report

The authors contribute an intelligent method with advanced technologies to solve road system generation in urban green space. This research is very interesting and enlightening. However, I believe several issues regarding the scientific justification and the organization of the manuscript must be addressed before this manuscript can be published in the International Journal of Environmental Research and Public Health (Please see my comments below)

1.     The current manuscript needs to be comprehensively revised to match the clarity required by a scientific paper, discarding a few redundant or less valuable portions, such as Figure. 2-5 should be removed. Also, please improve the visual quality of all figures to give readers a better understanding of the content.

2.     In the introduction, the authors did a good description of the importance of road system generation in urban green space. However, specifying a proper justification and a novelty description of the proposed method in the introduction seems weak. Why would the proposed method perform better than the others? What is your scientific justification for using a genetic algorithm? And what is the main idea behind the development of your strategy?

3.     The authors should add a new section, i.e., 4. Results and Discussion. Please reorganize sections 2 and 3, and move relevant content to section 4. The current version may be difficult for readers to understand.

Author Response

Comments1: The current manuscript needs to be comprehensively revised to match the clarity required by a scientific paper, discarding a few redundant or less valuable portions, such as Figure. 2-5 should be removed. Also, please improve the visual quality of all figures to give readers a better understanding of the content.

Response:

Thank you for your suggestion, we removed a few redundant or less valuable portions. Figure. 2-5 have been deleted. In addition, we considered the integrity of the graphic, unified the graphic, and some figures have been merged.

Specifically:

The original figure.16-18 was merged into the current figure.12(line549)

The original figure.19-22 was merged into the current figure.13(line568)

We reduced the number of graphics to make the text more coherent.

Comments2: In the introduction, the authors did a good description of the importance of road system generation in urban green space. However, specifying a proper justification and a novelty description of the proposed method in the introduction seems weak. Why would the proposed method perform better than the others? What is your scientific justification for using a genetic algorithm? And what is the main idea behind the development of your strategy?

Response:

Thank you for your point. In fact, the Chinese traditional garden is relatively "personalized", it was originally designed by literati, with cultural thoughts. The road generated by a genetic algorithm can simulate the genetic variation mechanism in nature to realize the optimal route design. It can solve complex path design problems and find the optimal solution.

And we also discussed its application effect in Chapter 4, which is a successful case applied in practical engineering. That's enough to say that it's also enough to support the application.

(line711-715)

Comments3: The authors should add a new section, i.e., 4. Results and Discussion. Please reorganize sections 2 and 3, and move relevant content to section 4. The current version may be difficult for readers to understand.

Response:

We restructured the article, and reorganized sections 3 and 4. Replace 3.4 with 4. Discussion and Results: Application effectiveness. (line683) Moreover, the limitation of application effect is added to the discussion in Chapter 4. (line711-715)

In addition, we have replaced Chapter 4 with Chapter 5 Conclusions. (line720)

Therefore, the structure of the paper is as follows:

  1. Introduction
  2. Materials and Methods
  3. Parametric construction and generation process of a garden road system
  4. Discussion and Results: Application effectiveness
  5. Conclusions

Reviewer 3 Report

The article focuses on a niche topic that I personally find interesting. I see a very strong emphasis on the practical application of research. However, the following revisions may be needed.

1.    The article is very hard to read. Its structure is cut by numerous graphics, which in my opinion are too many. As a result, the article is illegible and loses its continuity. In addition, the graphics are of very poor quality, visually worsening the visual perception of the overall content.

2.    You have described the method, analysis, etc. very precisely and extensively, but the results are not clear. Usually in scientific papers there is a separate section where you show your results - the reader should be able to find them quickly. I think that the problem with the readability of the results is the result of the lack of a clear and orderly structure of the entire article.

3.    As I mentioned, the article is very practical and less scientific. It has only 20 literature items. This is not a big problem, however, the very nature of the article requires that it agree with accepted standards in science. First, the article lacks scientific structure. You have created a "literature review" chapter and there is no discussion, which is a very important part of the publication. In the discussion, you should compare the items discussed in the research review with your own results (in other words, the discussion should include an interpretation of the research results and a confrontation of the obtained results with theories and the current state of research). Moreover, the discussion includes a limitation point. Each method used has limitations and these must be indicated.

Author Response

Comments1: The article is very hard to read. Its structure is cut by numerous graphics, which in my opinion are too many. As a result, the article is illegible and loses its continuity. In addition, the graphics are of very poor quality, visually worsening the visual perception of the overall content.

Response:

Thank you for your suggestion, we removed a few redundant or less valuable portions.  Figure. 2-5 have been deleted. In addition, we reduced the number of graphics to make the text more coherent. Some figures have been merged.

Specifically:

The original Figure.16-18 was merged into the current Figure.12(line549)

The original Figure.19-22 was merged into the current Figure.13(line568)

At the same time, we improved the quality of all the figures.

Comments2: You have described the method, analysis, etc. very precisely and extensively, but the results are not clear. Usually in scientific papers there is a separate section where you show your results - the reader should be able to find them quickly. I think that the problem with the readability of the results is the result of the lack of a clear and orderly structure of the entire article.

Response:

Thanks for your advice, we recognize the structure problem of the article. We restructured the article, and reorganized sections 3 and 4. Replace 3.4 with 4. Discussion and Results: Application effectiveness. (line683) Moreover, the limitation of application effect is added to the discussion in Chapter 4. (line711-715)

To make the structure of the article clearer, we have replaced Chapter 4 with Chapter 5 Conclusions. (line720)

Therefore, the structure of the paper is as follows:

  1. Introduction
  2. Materials and Methods
  3. Parametric construction and generation process of a garden road system
  4. Discussion and Results: Application effectiveness
  5. Conclusions

Comments3: As I mentioned, the article is very practical and less scientific. It has only 20 literature items. This is not a big problem, however, the very nature of the article requires that it agree with accepted standards in science. First, the article lacks scientific structure. You have created a "literature review" chapter and there is no discussion, which is a very important part of the publication. In the discussion, you should compare the items discussed in the research review with your own results (in other words, the discussion should include an interpretation of the research results and a confrontation of the obtained results with theories and the current state of research). Moreover, the discussion includes a limitation point. Each method used has limitations and these must be indicated.

Response:

Thank you for your comments, which are very helpful for our revision.  Our previous thinking perspective may indeed be narrow, only focusing on the study of the garden road of Chinese traditional gardens.

Gardens have developed into modern open spaces (parks) that can be used by the public and are a kind of urban green space.  Therefore, we added a review of sustainable urban green space design methods and related references of parametrized application research in 1.2 Literature review.

(line63-81;line114-166)

Round 2

Reviewer 2 Report

The revised edition met my expectations, and I would happily recommend publication in this edition.

Reviewer 3 Report

I accept the article after Authors’ corrections.